# Scholars as government-appointed research evaluators: Do they create congruence between their professional quality standards and political demands?

Hendrik Woiwode ◉ *

President's Research Group, Berlin Social Science Center, Berlin, Germany

* hendrik.woiwode@wzb.eu

## Abstract

All across the globe politically initiated research evaluations are based on "informed peer review" procedures. Scholars are appointed as evaluators and can apply self-defined quality standards in order to overcome shortcomings of standardized measures. Even though there are no binding criteria in these procedures and the quality standards of the scholars' disciplines vary, studies suggest that scholars, in their role as government-appointed research evaluators, assess research uniformly. By drawing on a small-N investigation, this study compares the quality standards scholars apply as government-appointed research evaluators with quality standards they follow as researchers. The study points to a paradox: Criteria scholars refer to while describing the excellence of their own research and criteria they use as evaluators differ and contradict each other. The results are discussed from different angles.

**Data Availability Statement:** All relevant data are within the manuscript and its Supporting Information files.

## Introduction

The mutual assessment of research-quality by qualified scientific peers—the so-called peer review—is the most widely accepted instrument for assessing research. It fulfills an elementary selection function within the scientific community as the review of manuscripts in the publication system, the awarding of academic prizes, or the review of research funding applications are based on peer review [1, 2]. However, peer review is not exclusively used for self-governing purposes within the scientific community. As the demand for control, accountability and evaluation of research performance has increased significantly in recent decades [3–5], peer review becomes increasingly used as a formalized management tool in politically initiated evaluations–mostly in combination with bibliometric information in so-called "informed peer review" procedures [6, 7].

Political initiated evaluations based on informed peer review–among them the Research Excellence Framework (REF) in the UK, or the Excellence in Research for Australia initiative (ERA)–were introduced with the aim to ensure an efficient allocation of scarce public funds and to stimulate better research performance. They influence structural and financial

**Funding:** The publication of this article was funded by the Open Access Fund of the Leibniz Association. The funders had no role in study design, data collection and analysis, decision to publish, or preparation of the manuscript.

**Competing interests:** The authors have declared that no competing interests exist.

conditions of academic knowledge production and are thus seen as a regulative pressure on professional self-regulation because they enforce politically defined quality standards of scientific work [8–10]. However,—due to the integration of a qualitative peer review mechanism -, active researchers review the quality of research during these evaluations. Taking over a role as evaluator enables scholars to enforce an understanding of quality that does justice to the special characteristics of their core activities that bibliometric data cannot reflect. Judgments about the (future) quality of research outputs are thus not made only based on citation information [7].

Definitions of high-quality research vary between (sub-)disciplines due to different epistemological orientations [11–13]. Standardized research evaluations are criticized because they focus only on specific output measures and thus cannot reflect this heterogeneity. Qualitative peer review procedures are therefore included in government-initiated evaluations. They should reflect the heterogeneity of scientific outputs and overcome the deficits of standardized evaluation procedures [7]. However, research suggests that in their role as government-appointed evaluators scholars from different disciplines emphasize the characteristics of high-quality research in nearly the same way–even though they are able to conduct the quality assessments themselves and without mandatory criteria [14]. This allows the assumption that–despite the integration of a qualitative peer review mechanism–the evaluation procedures do not reflect epistemic heterogeneity adequately and raises the question regarding the overall functionality of qualitative peer reviews in political initiated research evaluations.

To the best of the author's knowledge no study so far has systematically compared the quality standards scholars apply during political initiated qualitative research evaluations in so-called "informed peer review" procedures with quality standards they apply in their role as researchers. The majority of studies concerned with peer review focus on the assessment procedures in the publication sector and use quantitative methods to examine the procedures reliability, validity and fairness [15]. These studies give rise to criticism of peer review by indicating, for example, a low reliability [16, 17], a low prognostic quality [18] or opportunistic behavior of reviewers and authors [19]. The number of studies concerned with qualitative peer review in political initiated evaluations, however, is relatively small [20]. The present study thus asks:

Do scholars as government-appointed research evaluators create congruence between their professional quality standards and political demands?

The question is answered by focusing on the work of a federal state government agency in Germany who evaluated research performances of all subjects at universities in a German federal state between 1999 and 2007 on the basis of peer review procedures. The research question is investigated by means of a qualitative content analysis of semi-structured interviews with evaluators from three subjects with different epistemological cultures: history, electrical engineering and economics. After introducing the papers topic, the conceptual background of the study will be illustrated. Section three entails a presentation of the research context, methods, and data. In section four the findings are illustrated. The last section discusses the findings and concludes the paper.

## Background

Academic professionals are members of an "organized group that possesses esoteric knowledge which has economic value when applied to problems" [21]. Due to this esoteric knowledge and their core activities' high complexity they are traditionally equipped with high operational autonomy [22–24]. The New Public Management (NPM) reform agenda, however, has been changing academics working conditions. Until the late 20th century the

bureaucratic administration of universities in most OECD countries ensured a high degree of autonomy for scholars who could govern themselves within the "republic of science" [25, 26]. Due to NPM-reforms that took hold in the public sector of most OECD countries in the late 20th century, however, the fit between scholars' professional practices and environmental demands has become increasingly relevant [27]. NPM's core characteristics, i.e. competition, a private-sector management style, and output control mechanisms, reconfigured working in academia [28]. Accountability and output-oriented performance appraisals linking the allocation of resources to productivity indicators question professional autonomy by defining and enforcing external quality standards of academic performances—for example the publication of research findings in highly ranked journals or high amounts of third party funding [3, 5, 9, 29]. Scholars increasingly have to reach a congruence between their professional practices and external demands in order to preserve legitimacy, i.e. the generalized perception that their actions "are desirable, proper or appropriate within some socially constructed system of norms, values, beliefs and definitions" [30]. Taken together, the reform developments have put scholarly work and academic privileges "under pressure" [31] and are thus seen as a threat to professional self-determination [10, 32].

The increased relevance of political initiated research evaluations is a central component of the reform developments. Research evaluations that are solely based on research metrics like the journal impact factor indicating the average number of times articles in a journal have been cited in the previous years are widely criticized for not reflecting the epistemological heterogeneity of research outputs adequately [7]. In the arts and humanities, for example, bibliometric indicators are not considered as trustworthy and robust proxies of research output as in "the natural and the formal sciences, where publications in international journals and conference proceedings are the most accepted form for the diffusion of research outputs" [7]. In addition, research evaluations that assess the quality of publications based on the rating of the journal in which an article was published are criticized because the rating of a journal does not tell whether the singular article had actually a high scientific impact [5]. Furthermore, they are criticized for leading to unintended side effects like reactivity, meaning scholars change their behavior strategically—for example in order to reach success in terms of performance measures instead of sticking to their content-related research motivation [33–35]. In order to circumvent these deficits most political initiated evaluation procedures are structured as informed peer review exercises. These are characterized by qualitative expert reviews, which are informed by quantitative indicators such as citation information. Informed peer review procedures are intended to prevent assessments from being based solely on quantitative citation information and shall thus compensate for the deficits of quantitative measures [7].

It seems obvious that different epistemological styles are reflected during qualitative peer review in informed peer review exercises and common productivity indicators do not play a crucial role–at least in certain subjects. Against the background of different studies, however, it is not self-evident that scholars solely evaluate based on qualitative and discipline-specific criteria. Torka [36] investigated interpretations and judgment-patterns of scholars in scientific evaluation processes which took place during the British Research Assessment Exercise (RAE). He points to a high level of agreement concerning assessment criteria as the inspections was shaped by uniform and implicit standards. Examining the evaluation of research funding applications Langfeldt [37] observed an orientation towards a shared consensus of "good" research and noted the guidelines of the procedure hardly play a role for the reviewers as these let themselves guide by an unexplained "school of thought" [37]. Schiene and Schimank [14] examined the written reviews of the evaluation procedure focused in this study and found that reviewers from the humanities and social sciences emphasize the characteristics of good research in the same way as reviewers from the natural and engineering sciences. Despite the

heterogeneity of research practices, the same "recipes" of good research were represented in the reviews. Lamont [13] examined five interdisciplinary peer review panels in the USA that award grants and research funding. Lamont points out that reviewers of all disciplines refer to almost identical "scripts of excellence" (p. 170).

In the further course of this article it is thus explored whether scholars engaged as evaluators in political initiated research evaluations apply criteria that correspond to the quality standards they pursue in their role as researchers in order to create congruence between their professional standards and external evaluative pressures.

## Methods

The evaluation procedure of the institution focused on in this study is divided into four steps and based on a multi-level peer review procedure in which both quantitative and qualitative instruments are used. First, the subjects prepare a self-evaluation report presenting themselves and their research units based on quantitative measures. The self-evaluation report is forwarded to an expert group consisting of scholars of the discipline to be evaluated. These scholars are working in another state, are proposed by the institution and appointed by the federal states' Minister of Science. After receiving the report, the experts assess the research performance at the institutes during an on-site visit, usually lasting one day. This visit focuses on discussions with the scholars and the university management. In a third step, the assessments and recommendations are recorded in a confidential final report that, in a last step, is reviewed by the institution and handed over to the Ministry and the university [38, 39].

The institution mentioned a number of evaluation dimensions which should be considered by the reviewers in the official description of the procedure. These can be divided into the two areas of quality/relevance and effectiveness/efficiency. The mentioned criteria of the categories quality/relevance are: Innovativeness of research (scientific achievements in international comparison, reputation, new research frontiers), scientific impact (publications, conferences, third-party funding, etc.), interdisciplinarity or special status, cooperation at regional and national level, intensity of international cooperation, effectiveness of the promotion of young researchers, cooperation with industry, administration, politics and cultural institutions, and transfer. However, there is no mandatory operationalization of these criteria. It is the evaluators' responsibility to interpret the criteria. Concerning the dimension effectiveness/efficiency the description of the procedure's main features only states the evaluator should especially focus on the relationship between effort and success and regard whether the intended goals are being achieved with the resources in use (staffing, equipment, third-party funds from various sources) [14, 36, 38, 39]. The evaluators have a great leeway during the review process as the assessment dimensions are broad and described in a non-specific way.

### Research design & data collection

The data in this article are drawn from semi-structured, qualitative interviews with a total of 20 respondents. These include eleven interviews with respondents involved in the procedure as organizers, and a purposive sample of nine respondents involved in the evaluation process as evaluators. The author of this study aimed to select interviewees following different epistemological styles in order to ensure a maximum diversity of cases. Michele Lamont's [13] study has proven to be particularly suitable for systematizing differences in the evaluation of scientific excellence. Lamont identifies four different epistemological styles capturing different evaluative and epistemic cultures: a comprehensive style characterized by an appreciation of "comprehensive" research, a focus on detail, and consideration of the specificity of the research context, a constructivist style characterized by an appreciation of reflexivity and consideration

of the researchers own identity-based perspective, a positivist style characterized by an appreciation of the generalizability of research results and hypothesis-checking procedures and a utilitarian style exclusively valuing the production of instrumental knowledge. As these "sets of conventions" (p. 54) contain ideas about the correct way of collecting data and the function of theories in the research process, they influence how excellence of research is defined [13].

Lamont notes that economics and history are the two disciplines within which scholars have the highest consensus on what constitutes high-quality research in their discipline. She further points out that humanities scholars consider interpretative skills to be essential for the production of high-quality science, whereas the empirically oriented social sciences tend to view interpretation as a destructive force. This difference influences the evaluation of research [13]. Economics can be assigned to the empirically oriented social sciences and history to the humanities, or the "understanding" social sciences. Therefore, the author of this study interviewed three evaluators from history and three from the field of economics. In order to cover a discipline which, in contrast to the other two, is characterized by a high degree of application relevance, the author also interviewed three evaluators from electrical engineering. The technical sciences can be characterized as purposeful, functional, pragmatic and effective [11].

Interviewees were first asked concerning their motivation for participating voluntarily in political initiated evaluations as evaluators and their general attitude towards political initiated evaluations by asking questions like "Why did you agree to participate in the evaluations as an evaluator?" or "What is your basic opinion of politically initiated evaluations of scientific achievements?". Afterwards the questions focused their perspective on excellence in their role as researchers by asking questions like "What is your most excellent research work?" or "What characterizes the excellence of this research work?". The following block focused on the interviewees' role as evaluator in politically initiated evaluations by asking questions like "How do you define academic excellence in your role as an evaluator during the political initiated evaluation?".

Each interview lasted between 40 and 90 minutes. The author stopped gathering interview material when saturation occurred, i.e. when newly collected data were redundant [40, 41]. The interviews were conducted with an interview guideline and coded using MAXQDA 20.

The analysis of the collected material was based on a qualitative content analysis [42]. Due to the research questions' narrow scope, the categories were not derived from the material, but from the assumptions guiding the structure of the interview guideline. The deductive categories were thus systematized using the thematic blocks of the interview guide. The author took care that the category system consists of precisely defined categories. After the transcripts had been read intensively and important text passages had been marked, the interview responses were filtered on the basis of previously formed categories [43]. After the material was coded, all text passages with the same main category were bundled and compared with each other. The general deductive main categories were differentiated after the first coding process. Some inductive subcategories were added. After the category system was differentiated, the entire interview material was coded using the new category system [41, 42]. Fig 1 visualizes the steps of the data analysis and the category system. In order to avoid a researcher perception bias, the author discussed his categorizations with colleagues from his field of research. Two other coders checked coded extracts in order to minimize the source of error of subjective interpretations and to ensure that each theme captures the data accurately (intercoder agreement). The focus of this procedure was on the coding qualities' practical improvement. The author thus didn't focus on the percentage of agreement or the coefficient. Instead, he aimed to address code assignments that do not match in order to work with accurately coded material [44, 45]. In addition, the first results of the data analysis–i.e. a preliminary typology–were discussed during a group discussion.

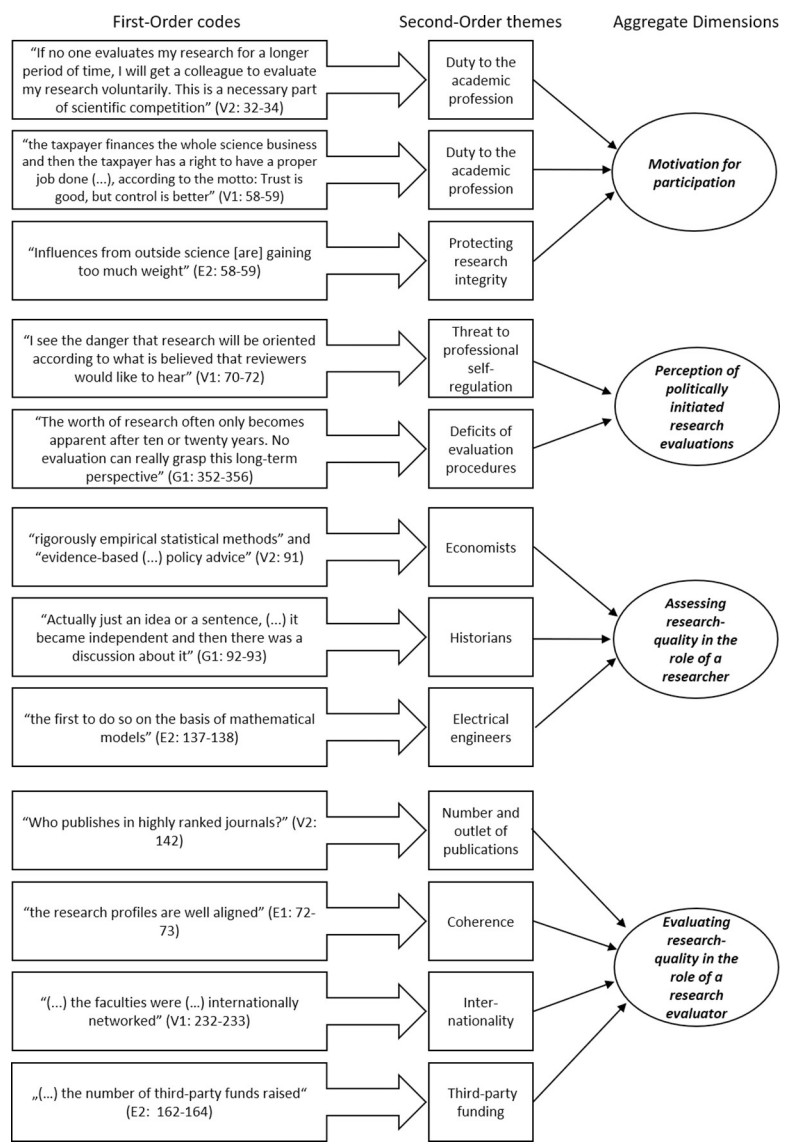

**Fig 1. Data analysis.**

## Results

In the following section, the relevant text passages of the main thematic categories are presented in descriptive form. The presentation follows the thematic blocks of the interview guideline.

### Motivation for participation: Evaluating as part of the professional role

The data suggest that the motivation for participating in government-initiated research evaluations is predominantly driven by three central aspects: a perceived duty to the academic profession, a perceived duty to the public, and a will to protect the integrity of research.

**Duty to the academic profession.** The interviewed scholars consider participation in evaluations as a part of scientific self-administration and as a duty towards the academic profession. Interviewee G2 emphasizes that participation in evaluations "is part of the professional

duties of a university teacher" (G2: 261). The evaluation of scientific achievements is a work "which links elements of scientific discourse with elements of self-administration" (G2: 34–35). The "critical evaluation of what one presents as a scientist" (G2: 32) is "a basic component of the scientific discussion" (G2: 32–33) and academic self-administration is "a part of the professional task" (G2: 41–42). Interviewee V2 states that evaluations conducted by academic colleagues are "part of a simulation of competition" (V2: 71) and necessary to "make us better" (V2: 68–69). He also considers the regular evaluation of his own performance as a necessity: "If no one evaluates my research for a longer period of time, I will get a colleague to evaluate my research voluntarily. This is a necessary part of scientific competition" (V2: 32–34). In addition, he states that he performs the activity on the basis of a "commitment to discipline" (V2: 227). V1 also regards evaluations as a commitment to his colleagues when he says: "I also use peer review services" (V1: 36–37).

**Duty to the public.**   The interviewees see it as their task to account to the public for the actions of scholars on behalf of science. Interviewee V2 states that science is "part of society" (V2: 68) and "it is important to make the best out of the resources one (. . .) gets" (V2: 72–73). Interviewee V1 states "the taxpayer finances the whole science business and then the taxpayer has a right to have a proper job done (. . .), according to the motto: Trust is good, but control is better" (V1: 58–59). He emphasizes the positive effects of the transparency generated by evaluations: "In some cases civil servants' research performance can be improved—to put it mildly" (V1: 59–63).

**Protecting research integrity.**   The interviewees stress they do not accept externally defined criteria of high research quality and claim to be self-determined in judging research performance. Taking over a role as research evaluator is seen as a way of preventing non-scientific actors, especially governmental ones, from exerting too much influence on the shaping of the scientific field and as a way to prevent evaluations from being carried out "by pure science managers" (E2: 36–37), "who (. . .) do not even know what research really is in today's world" (E2: 44–48). The interviewees stress that only qualified scholars should carry out research evaluations. Acting as an evaluator during politically initiated evaluations prevents "influences from outside science gaining too much weight" (E2: 58–59).

## Perception of politically initiated research evaluations

The data suggest that the interviewees are critical of the growing relevance of politically initiated research evaluations. This criticism is driven by two central aspects: a perceived threat to professional self-regulation and deficits in evaluation procedures.

**Threat to professional self-regulation.**   Scholars perceive politically initiated evaluations as a threat to their professions' core activities and their self-regulation. All interviewees expressed criticism of the consequences and the increasing relevance of evaluation procedures. Economist V1 points out evaluations may prevent innovative research approaches, since the evaluated researchers may be guided by reviewers' preferences, stating "I see the danger that research will be oriented according to what is believed that reviewers would like to hear" (V1: 70–72). Interviewees E1 and E2 stress that innovative approaches may get lost through the evaluation process stating "Really revolutionary new ideas could of course get lost, because they have perhaps not yet been published or there is no one on the evaluation committee who recognizes their potential" (E2: 252–254); or "It is difficult to apply research funds with unconventional proposals, because other scholars read the proposal and say: "This is impossible"(E1: 165–168).

**Deficits of evaluation procedures.**   Economist V2 criticizes the superficial character of evaluations: "Every evaluation is a very superficial observation of what is happening (. . .) It

never does justice to particularities" (V2: 253–254). Historian G2 also stresses "there is certainly much left over that even the attentive evaluator can miss" (G2: 197) and that "an evaluator must be aware of the fact that he cannot look into all peculiarities and niches of one research-area" (G2: 198–200). Historian G1 points out that the status quo observed in evaluations does not necessarily mirror the true value of the research evaluated: "The worth of research often only becomes apparent after ten or twenty years. No evaluation can really grasp this long-term perspective" (G1: 352–356). Furthermore, he is critical of metrics for measuring research: "At first, it does not matter whether I have one hundred or one hundred and fifty publications. The important question is: What was the starting point of these one hundred and fifty or one hundred publications? And: How are they discussed in the field? Are they still discussed?" (G1: 357–360). With the exception of one interviewee, all of the interviewed evaluators share this critical stance towards output indicators for measuring research, since these are objectified by external actors and based on quantifying methods.

## Assessing research-quality in the role of a researcher

Descriptions of the excellence in the role of a researcher are nearly identical within the disciplines, but differ between disciplines.

**Economists.** Both economists state that their excellent work is characterized on the one hand by the application of innovative econometric techniques and on the other hand by economic policy implications. Asked about his most excellent research interviewee V2 elaborates on his work on migration research, which has shown that "ethnic diversity is good for the economy" (V2: 83) and that migration "is not harmful to the host society" (V2: 84). This work, he further elaborates, is based on "rigorously empirical statistical methods" and delivers "evidence-based (. . .) policy advice" (V2: 91). Interviewee V1 refers to his work on structural unemployment characterized by "theory, proper empirical verification and then (. . .) economic policy implications" (V1: 146–148). He points out the work is excellent although it appeared in the German Economic Review, "which is not an A-Journal" (V1: 150).

**Historians.** The historians state that their excellent work has coined terms or theses that have triggered a change in the way of thinking both within the specialist community and beyond the specialist community. Historian G1 states that his most excellence work was "Actually just an idea or a sentence, (. . .) it became independent and then there was a discussion about it" (G1: 92–93). Asked which of his works he considers particularly excellent, Historian G2 refers to his work in the field of social history distinguished by the fact that it has established a new, "very broad understanding of social history" (G2: 66) and "strongly linked historical scholarship and historically oriented social sciences" (G2: 68–69). Moreover, the "broad comparisons of Germany in international comparison (. . .) is a second characteristic" (G2: 72–73) of his work. Historian G1 indirectly criticizes a strong focus on output indicators stating "It can take years before a groundbreaking discovery reaches the scientific community. Until then, scholars can be virtually invisible and publish little or nothing" (G1: 101).

**Electrical engineers.** The electrical engineers state their excellent work is characterized by an application of innovative calculation models and emphasize that research is excellent when it has led to something "about which the whole scientific world, including reviewers, have initially said: This cannot work" (E1: 155–157). Interviewee E2 refers to his work in the field of radiation protection and optimization of cancer treatments characterized by the application of innovative methods. He and his colleagues were "the first to do so on the basis of mathematical models" (E2: 137–138). Taken together, the engineers rated their research as excellent, if it resulted in a previously unknown solutions to a problem as well as to new technical applications.

### Evaluating research quality in the role of a research evaluator

When describing the characteristics of high-quality research in the context of research evaluations, the evaluators of all three disciplines refer to similar criteria. There are no significant subject-specific differences with regard to the criteria applied. All interviewed evaluators refer to similar productivity indicators.

**Number and outlet of publications.**   Reviewers from all three disciplines state that they are mainly guided by the number of publications and–except for historians–journal rankings. Interviewees state that the decisive factor is the "number of publications in the field" (E2: 162–164). Evaluations always focus on the question: "Is (research) internationally visible? Do the colleagues publish in the highly ranked journals?" (E2: 204–205); or, "It is always about: Who publishes in highly ranked journals?" (V2: 142). The historians also refer primarily to the publications stating that there are always texts "which can be scaled in some way compared to others" (G1: 205–206). Anyone who "does not write any books in history, in my view, has certain shortcomings" (G1: 301–303). Both electrical engineers say that international visibility always plays a role. E1 emphasizes that international visibility always plays a role, "that is (. . .) how far things have been taken up internationally by other scholars, influenced their work and the like" (E1: 225–228).

**Coherence.**   Reviewers from all three disciplines state that they were paying attention to the coherence of the research profile of the evaluated institutes, emphasizing, for example, that it matters whether an institute had "a coherent program" (V2: 149) as it is a sign of quality if "there is some kind of center formation, so that one can also achieve something as a group" (V2: 150–151), if scholars were able to "present themselves in a certain profile" (G1: 163–164), or, that it was relevant whether "the research profiles are well aligned" (E1: 72–73), because it "makes no sense if everyone does exactly the same thing" (E1: 73–74).

**Internationality.**   Reviewers in all three disciplines refer to internationality stressing, for example, that it is a sign of quality if the scholars evaluated "have worked with (. . .) researchers in Cambridge or Harvard or something like that" (G1: 320–323), that it always plays a role, whether researchers have "appropriate international contacts" (E2: 205), or "care was taken to ensure that the faculties were then also internationally networked" (V1: 232–233).

**Third-party funding.**   Interviewees emphasize the importance of third-party funding, which plays a role even though it is "more of a function derived from something else" (V2: 152–153). In electrical engineering, spin-offs of companies and patents play a role.

## Discussion & conclusion

This article asked whether scholars as government-appointed research evaluators create congruence between their professional quality standards and political demands. After introducing the paper's topic, the author illustrated the conceptual background of the study. Section three entailed a presentation of context, methods, and data. In section four, the findings were outlined.

The findings suggest that, first, the interviewed scholars regard their voluntary participation in evaluations as a part of scientific self-administration and as a duty towards the academic profession. When evaluating their own research concerning its excellence, scholars place emphasis on content-related innovations–for example, specific combination of research methods or theories. Research is described as excellent despite being published in a journal "which is not an A-Journal" (V1: 150). The interview data suggest, however, that such content-related aspects do not play a key role during research evaluations. In their role as evaluators scholars seem to be mainly guided by dominant output indicators, e.g. publications in highly ranked journals or third-party funds, as well as by "throughput" criteria such as the coherence of the

faculty being evaluated. Scholars from all disciplines emphasize to be critical concerning the implications of dominant output indicators, and stress that their usage may promote opportunistic behavior and inhibit innovation and that dominant research metrics have a low prognostic value. They criticize the fact that the number of publications in high-ranking scientific journals has become the leading currency of scientific reputation and point to the deficits and risks of this development, which are illustrated by various studies and treatises. In this stance they argue in line with the Declaration on Research Assessment (DORA) signed by leading scientific institutions, according to which the assessment of scientific quality of individual contributions, promotion, hiring, and funding decisions should not solely be based on research metrics like the impact factor [46]. The data suggest, however, that while assessing research quality in their role as evaluators, they mainly draw on dominant metrics. In the interview with interviewee E1, this ambivalence becomes particularly apparent. First of all, he emphasizes that, from his perspective as a researcher, research is excellent when it has led to something "about which the whole scientific world, including reviewers, have initially said: This cannot work" (E1: 155–157). He points to the difficulty of gaining innovative insights as it is hardly possible to get unconventional research proposals approved. "These proposals are read by other scholars who say: This is impossible" (E1: 167–168). Describing excellent research in the context of the research evaluation, however, interviewee E1 refers to dominant output indicators, e.g. the ranking of the journals published in, the number of third-party funds raised, or the spin-offs at the evaluated site. He does not elaborate on specific content-related attributes. As a researcher, E1 states to consider it desirable to use innovative research approaches, to reject common views of problems and to establish new ways of solving familiar problems that contradict dominant paradigms. He criticizes dominant metrics for not capturing the value of research with these attributes adequately or even impeding research with these attributes. The data suggest, however, that in the evaluation process he nevertheless applies criteria that he thinks would prevent innovative research approaches and rather promote "mainstream research" (E1: 173).

How can the discrepancy be explained? Scholars are being asked to evaluate the potential for future research productivity. The evaluation process involves prediction as the value of current research is not known at the time the evaluation is conducted. The benefits of scientific research often become apparent after a long period of time. In medicine, for example, it takes on average three decades to translate basic research into practical applications [47]. Given the limited time available for judging research performance, metrics of past research outputs may appear to evaluators to be the most or only reliable and efficient instruments for predicting the value and potential success of future research outputs. Qualitative properties of the research evaluated are not accessible within the given period of time. The scholars interviewed have the benefit of retrospection in assessing the quality of their own research. In their role as evaluators, however, they have to predict the future value of research. The evaluators thus seem to find themselves in a dilemma having to use instruments indicating the quality of past research outputs, whose accuracy in terms of determining future research quality they call into question.

There are doubts about the effectiveness of peer review procedures that are based on the disadvantages the procedure has compared to bibliometric methods. Critics of the peer review process stress the better reliability of bibliometric forms of assessing scientific output and point out that it is impossible, for reasons of time and cost alone, to (continuously) assess the research performance of a scientific system using peer review–here too, they see bibliometric methods as having a clear advantage. Critics also point out that the efficiency of a (scientific) production system cannot be determined by qualitative assessment procedures alone—this requires quantifying productivity indicators. Other criticisms are that peer review–in

combination with output measures–jeopardizes a measurement systems' robustness, and hampers the measures validity. In summary, robustness, validity, functionality, and cost and time effectiveness are seen as advantages of research evaluation based solely on bibliometric evaluation methods (see Abramoa and D'Angelo [7] for an overview). However, abolishing the peer review mechanism in politically initiated evaluations might be interpreted as an attack on academics' professional self-regulation privileges. Research demonstrates that entities sometimes "respond to social pressures by superficially conforming in order to appear legitimate to external audiences as a means of buffering and protecting their core economic or technical activities" [48]. The results suggest that appointed scholars are motivated to preserve the legitimacy of professional self-governance, the main instrument of self-control of the academic profession, and to prevent nonscientific actors, especially governmental ones, from exerting too much influence on the scientific field. Whether it is actually possible to capture the quality of research in the context of qualitative evaluations seems to play a minor role. This might be the reason that as government-appointed research evaluators scholars indirectly foster policy conditions that contradict the ideals expressed in their role as researchers. From this perspective the findings may thus be seen as an expression of a paradox. The findings suggest that as government-appointed research evaluators scholars do not create congruence between their professional quality standards and political demands.

The study has several limitations that create necessity for further research. The sample size was large enough to sufficiently explore the phenomenon of interest and to address the research question. The study followed the methodological principle of saturation. Saturation occurred as adding more participants to the study would not have resulted in additional perspectives or information regarding the papers core topic, i.e. the exploration whether scholars as government-appointed research evaluators create congruence between their professional quality standards and political demands. However, due to the qualitative small-N study design the findings are limited in terms of their breadth and scope. One major disadvantage of purposive samples like the one used in this study is that they can be prone to a researcher bias. The judgmental, subjective component of purposive sampling is a weakness as the sample has been based on the researcher's judgment. Compared to probability sampling techniques designed to reduce researcher biases this is a clear disadvantage. However, the judgment concerning the purposive sample drawn on in this study has been based on clear theory-driven criteria that were illustrated in the data analysis section. The small N design and purposive sampling procedure have clear deficits concerning theoretical, logical, or analytical generalization. It's worth investigating whether selecting different units leads to different results [49]. Further studies could survey more evaluators from additional disciplines who took part in informed peer review procedures and validate the findings by using probability sampling techniques. In addition, it is worth investigating how structural changes in evaluation procedures, such as the time available for review, affect the behavior of reviewers and especially the core results of this study. This could provide policy makers with additional helpful information regarding the design of evaluation procedures.

## Supporting information

**S1 Table.**
(DOCX)

## Author Contributions

**Conceptualization:** Hendrik Woiwode.

**Data curation:** Hendrik Woiwode.

**Formal analysis:** Hendrik Woiwode.

**Investigation:** Hendrik Woiwode.

**Methodology:** Hendrik Woiwode.

**Project administration:** Hendrik Woiwode.

**Resources:** Hendrik Woiwode.

**Software:** Hendrik Woiwode.

**Supervision:** Hendrik Woiwode.

**Validation:** Hendrik Woiwode.

**Visualization:** Hendrik Woiwode.

**Writing – original draft:** Hendrik Woiwode.

**Writing – review & editing:** Hendrik Woiwode.

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
