## [Decision Letter · Decision Letter 0]

10 Jul 2020

PONE-D-20-07099

Scholars as government-appointed research evaluators: Do they create congruence between their professional quality standards and political demands?

PLOS ONE

Dear Dr. Woiwode,

Thank you for submitting your manuscript to PLOS ONE. After careful consideration, we feel that it has merit but does not fully meet PLOS ONE’s publication criteria as it currently stands. Therefore, we invite you to submit a revised version of the manuscript that addresses the points raised during the review process.

We look forward to receiving your revised manuscript.

Kind regards,

Cindy Sing-bik Ngai

Academic Editor

PLOS ONE

Journal Requirements:

Reviewers' comments:

Reviewer's Responses to Questions

**Comments to the Author**

1. Is the manuscript technically sound, and do the data support the conclusions?

Reviewer #1: Partly

Reviewer #2: Partly

2. Has the statistical analysis been performed appropriately and rigorously? 

Reviewer #1: N/A

Reviewer #2: N/A

3. Have the authors made all data underlying the findings in their manuscript fully available?

Reviewer #1: No

Reviewer #2: Yes

4. Is the manuscript presented in an intelligible fashion and written in standard English?

Reviewer #1: Yes

Reviewer #2: Yes

5. Review Comments to the Author

Reviewer #1: I found this manuscript to be very interesting. I think there is a clear difference between what scholars say they do (or intend to do) in peer review situations and what actually happens. It is interesting and useful to point out this paradox. However, I think there are reasons to expect this to be the case which should be addressed, and I also think that the authors overreach with some of their conclusions.

First, I was struck by the challenge of evaluating the potential for high-quality research with qualitative, discipline-specific criteria. From what I could gather from the description of the study (not being familiar with the evaluations myself), the process being examined is for continued funding of institutes that conduct research. The scholars were tasked with evaluating the institute's activities and recommending funding (if this impression is incorrect then further description is required in the paper). This means that scholars are being asked to evaluate the potential for future research and productivity, which is a difficult assignment. The value of future research, or even current research that has just been conducted, is largely unknowable (comments from E2 on p. 11 support this). The authors make much of the differences between the respondents' assessments of their own quality research and how they evaluate in the examined process. However, I question this disconnect for two reasons. First, the answers about their own research are somewhat subjective. Not only do the respondents know their own work the best, but they also have the benefit of retrospection in assessing quality. Second, as noted by G1 (p.12), the real value of research is not usually known for quite some time. So evaluating a research plan for an institute involves a considerable amount of prediction. The big question then becomes, what criteria are relevant or useful to evaluate potential success? Here standard markers and metrics seem logical. It is true that the respondents argue in favour of peer review and their ability to evaluate better than non-scientific actors. In the realm of judging good research from bad research this makes sense. But when evaluating research activity that will (hopefully) yield high-quality future outputs, how practical is this? I would have liked to see the authors address the very real difference between past and future research outputs, as well as probe further into how the respondents felt about the review process and what they were tasked with doing overall.

Second, I think that the authors overreach with their conclusions in lines 417-423. I am not convinced that researchers apply the evaluation standards they do to present academia in a specific way or that they are aware of the differences between their standards of quality research and their evaluation criteria. If there is further evidence that supports these contentions it should be added to the manuscript.

In sum, I think this research has value for assessing the utility of peer review but I would like to see the authors address some of the issues I have raised above.

I could not find supplementary data files to assess the availability of the data.

Reviewer #2: Introduction

On page 2 – line 40-41, it is mentioned that “…the demand for control, accountability and evaluation of research performance has increased significantly…” and on page 5 – line 111, “…allocation of resources is increasingly linked to productivity indicators.” Can more published literature be drawn on to support these claims? For example, please see Burrows, R. (2012) ‘Living with the h-index? Metric assemblages in the contemporary academy’, The Sociological Review, 60(2), pp.355–372.

Line 126 – “indicators don not play a…” should be “indicators do…”

Methods

The sample size of nine for the evaluators (coming from a limited range of disciplines: history, economics and electrical engineering) is very small.

The method of analysis needs to be explained in more detail – lines 217-224.

Were the themes reviewed and carefully checked by another coder to ensure that each theme accurately captured the coded extracts and the data set? You need to provide more details.

Results

The findings are illuminating but it is worth further categorizing the themes and elaborating on them as it appears that there are a number of subsidiary aspects to each theme.

For example, for the theme of Motivation for participation: Evaluating as part of the professional role (line 229), the subsidiary aspects are:

• Duty to the academic profession

• Duty to the community

• Protecting research integrity

The same can be applied to other themes to provide more clarity to the reader.

Discussion and Conclusion

Is there any published literature to support the conclusions of the study – lines 410-423?

Line 422 - …it should be “research performance…” rather than performances.

Line 427 – “unconventional research approaches that not succeed in terms of…”

should be “unconventional research approaches that do not succeed in terms of…” Please revise the language accordingly.

More practical and theoretical implications of the findings should be given.

The limitations of this study should be elaborated on, with the main one being the small sample size and that qualitative studies have limitations in terms of their breath and scope.

More suggestions for further research should be provided.

6. PLOS authors have the option to publish the peer review history of their article (what does this mean?). If published, this will include your full peer review and any attached files.

Reviewer #1: No

Reviewer #2: No

---

## [Author Response · Author response to Decision Letter 0]

1 Sep 2020

Response to Reviewers

 August 30, 2020

Ref: PONE-D-20-07099

Scholars as government-appointed research evaluators: Do they create congruence between their professional quality standards and political demands?

Journal: PLOS ONE

Dear Cindy Sing-bik Ngai,

thank you for your email dated July 10, 2020 enclosing the reviewer’s comments. I have considered all comments and revised the document accordingly. In the following I respond to each point raised in the section “Review Comments to the Author” and describe my corresponding revisions.

I hope that the revised version of the manuscript is suitable for publication in PLOS ONE and look forward to hear from you soon.

Sincerely yours,

Hendrik Woiwode

-- 

Dr. Hendrik Woiwode

Research Fellow

Group of the President

WZB Berlin Social Science Center

10785 Berlin

Tel: ++49-30-25491-534

Fax: ++49-30-25491-530

Geschäftsführung:

Prof. Jutta Allmendinger Ph.D./

Ursula Noack

Sitz der Gesellschaft Berlin

Amtsgericht Charlottenburg

HRB 4303

 

Reviewer #1

Reviewer #1 major comment 1:

“First, I was struck by the challenge of evaluating the potential for high-quality research with qualitative, discipline-specific criteria. From what I could gather from the description of the study (not being familiar with the evaluations myself), the process being examined is for continued funding of institutes that conduct research. The scholars were tasked with evaluating the institute's activities and recommending funding (if this impression is incorrect then further description is required in the paper). This means that scholars are being asked to evaluate the potential for future research and productivity, which is a difficult assignment. The value of future research, or even current research that has just been conducted, is largely unknowable (comments from E2 on p. 11 support this). The authors make much of the differences between the respondents' assessments of their own quality research and how they evaluate in the examined process. However, I question this disconnect for two reasons. First, the answers about their own research are somewhat subjective. Not only do the respondents know their own work the best, but they also have the benefit of retrospection in assessing quality. Second, as noted by G1 (p.12), the real value of research is not usually known for quite some time. So evaluating a research plan for an institute involves a considerable amount of prediction. The big question then becomes, what criteria are relevant or useful to evaluate potential success? Here standard markers and metrics seem logical. It is true that the respondents argue in favour of peer review and their ability to evaluate better than non-scientific actors. In the realm of judging good research from bad research this makes sense. But when evaluating research activity that will (hopefully) yield high-quality future outputs, how practical is this? I would have liked to see the authors address the very real difference between past and future research outputs, as well as probe further into how the respondents felt about the review process and what they were tasked with doing overall.”

Authors’ response to reviewer #1 major comment 1:

Thank you very much for your comment. I agree to the points mentioned. However, it somehow seems I have not considered these aspects clearly in the papers line of argumentation so far. I revised the text in order to address the points you mentioned - especially the difference between past and future research outputs. I have improved the text as shown below:

“Scholars are being asked to evaluate the potential for future research productivity. The evaluation process involves prediction as the value of current research is not known at the time the evaluation is conducted. The benefits of scientific research often become apparent after a long period of time. In medicine, for example, it takes on average three decades to translate basic research into practical applications [48]. Given the limited time available for judging research performance, metrics of past research outputs may appear to evaluators to be the most or only reliable and efficient instruments for predicting the value and potential success of future research outputs. Qualitative properties of the research evaluated are not accessible within the given period of time. The scholars interviewed have the benefit of retrospection in assessing the quality of their own research. In their role as evaluators, however, they have to predict the future value of research. The evaluators thus seem to find themselves in a dilemma having to use instruments indicating the quality of past research outputs, whose accuracy in terms of determining future research quality they call into question.

There are doubts about the effectiveness of peer review procedures that are based on the disadvantages the procedure has compared to bibliometric methods. Critics of the peer review process stress the better reliability of bibliometric forms of assessing scientific output and point out that it is impossible, for reasons of time and cost alone, to (continuously) assess the research performance of a scientific system using peer review – here too, they see bibliometric methods as having a clear advantage. Critics also point out that the efficiency of a (scientific) production system cannot be determined by qualitative assessment procedures alone - this requires quantifying productivity indicators. Other criticisms are that peer review – in combination with output measures - jeopardizes a measurement systems’ robustness, and hampers the measures validity. In summary, robustness, validity, functionality, and cost and time effectiveness are seen as advantages of research evaluation based solely on bibliometric evaluation methods (see Abramoa and D’Angelo [7] for an overview). However, abolishing the peer review mechanism in politically initiated evaluations might be interpreted as an attack on academics’ professional self-regulation privileges. Research demonstrates that entities sometimes “respond to social pressures by superficially conforming in order to appear legitimate to external audiences as a means of buffering and protecting their core economic or technical activities” [33]. The results suggest that appointed scholars are motivated to preserve the legitimacy of professional self-governance, the main instrument of self-control of the academic profession, and to prevent nonscientific actors, especially governmental ones, from exerting too much influence on the scientific field. Whether it is actually possible to capture the quality of research in the context of qualitative evaluations seems to play a minor role. This might be the reason that as government-appointed research evaluators scholars indirectly foster policy conditions that contradict the ideals expressed in their role as researchers. From this perspective the findings may thus be seen as an expression of a paradox. The findings suggest that as government-appointed research evaluators scholars do not create congruence between their professional quality standards and political demands.” (428-466)

Reviewer #1 major comment 2:

“Second, I think that the authors overreach with their conclusions in lines 417-423. I am not convinced that researchers apply the evaluation standards they do to present academia in a specific way or that they are aware of the differences between their standards of quality research and their evaluation criteria. If there is further evidence that supports these contentions it should be added to the manuscript.”

Authors’ response to reviewer #1 major comment 2:

Thank you very much for your comment. I agree to the points mentioned. I got rid of the argumentation that researchers apply the evaluation standards they do to present academia in a specific way or that they are aware of the differences between their standards of quality research and their evaluation criteria. I just wanted to discuss this aspect as a possible perspective. However, in accordance with the other peers comments it somehow seems I have indeed overreached with this conclusion. I have improved the text as shown below:

“Critics also point out that the efficiency of a (scientific) production system cannot be determined by qualitative assessment procedures alone - this requires quantifying productivity indicators. Other criticisms are that peer review – in combination with output measures –jeopardizes a measurement systems’ robustness, and hampers the measures validity. In summary, robustness, validity, functionality, and cost and time effectiveness are seen as advantages of research evaluation based solely on bibliometric evaluation methods (see Abramoa and D’Angelo [7] for an overview). However, abolishing the peer review mechanism in politically initiated evaluations might be interpreted as an attack on academics’ professional self-regulation privileges. Research demonstrates that entities sometimes “respond to social pressures by superficially conforming in order to appear legitimate to external audiences as a means of buffering and protecting their core economic or technical activities” [33]. The results suggest that appointed scholars are motivated to preserve the legitimacy of professional self-governance, the main instrument of self-control of the academic profession, and to prevent nonscientific actors, especially governmental ones, from exerting too much influence on the scientific field. Whether it is actually possible to capture the quality of research in the context of qualitative evaluations seems to play a minor role. This might be the reason that as government-appointed research evaluators scholars indirectly foster policy conditions that contradict the ideals expressed in their role as researchers. From this perspective the findings may thus be seen as an expression of a paradox.” (446-464)  

Reviewer #2 

Reviewer #2 comment on Introduction

“On page 2 – line 40-41, it is mentioned that “…the demand for control, accountability and evaluation of research performance has increased significantly…” and on page 5 – line 111, “…allocation of resources is increasingly linked to productivity indicators.” Can more published literature be drawn on to support these claims? For example, please see Burrows, R. (2012) ‘Living with the h-index? Metric assemblages in the contemporary academy’, The Sociological Review, 60(2), pp.355–372.”

Authors’ response:

Thank you very much for your useful hints. I drew on more literature at the places you mentioned in order to support the claims. The following literature is additionally drawn on in the revised version:

Burrows R. Living with the H-Index? Metric Assemblages in the Contemporary Academy. The Sociological Review. 2012; 60:355–72. doi: 10.1111/j.1467-954X.2012.02077.x.

Osterloh M, Frey BS. Ranking games. Eval Rev. 2015; 39:102–29. doi: 10.1177/0193841X14524957 PMID: 25092865.

Osterloh M. Governance by Numbers. Does It Really Work in Research? Analyse & Kritik. 2010; 32:267–83. doi: 10.1515/auk-2010-0205.

“Line 126 – “indicators don not play a…” should be “indicators do…””

Authors’ response:

Thank you very much for this hint. I apologize for my inattention. I have revised the line according to your remark.. 

Reviewer #2 comment on Methods

“The sample size of nine for the evaluators (coming from a limited range of disciplines: history, economics and electrical engineering) is very small.

The method of analysis needs to be explained in more detail – lines 217-224.

Were the themes reviewed and carefully checked by another coder to ensure that each theme accurately captured the coded extracts and the data set? You need to provide more details.”

Authors’ response:

Thank you very much for this useful comment. I have significantly expanded the methods section and described the data analysis procedure in more detail. I have also referred to the reason for the small sample size and the limitations that go along with it. Additionally, I have added a graphic that illustrates the data analysis. I have improved the text as shown below:

“The analysis of the collected material was based on a qualitative content analysis [43]. Due to the research questions’ narrow scope, the categories were not derived from the material, but from the assumptions guiding the structure of the interview guideline. The deductive categories were thus systematized using the thematic blocks of the interview guide. The author took care that the category system consists of precisely defined categories. After the transcripts had been read intensively and important text passages had been marked, the interview responses were filtered on the basis of previously formed categories [44]. After the material was coded, all text passages with the same main category were bundled and compared with each other. The general deductive main categories were differentiated after the first coding process. Some inductive subcategories were added. After the category system was differentiated, the entire interview material was coded using the new category system [42,43]. Fig 1 visualizes the steps of the data analysis and the category system. In order to avoid a researcher perception bias, the author discussed his categorizations with colleagues from his field of research. Two other coders checked coded extracts in order to minimize the source of error of subjective interpretations and to ensure that each theme captures the data accurately (intercoder agreement). The focus of this procedure was on the coding qualities’ practical improvement. The author thus didn’t focus on the percentage of agreement or the coefficient. Instead, he aimed to address code assignments that do not match in order to work with accurately coded material [45,46]. In addition, the first results of the data analysis – i.e. a preliminary typology – were discussed during a group discussion.” (219-238)

“The study has several limitations that create necessity for further research. The sample size was large enough to sufficiently explore the phenomenon of interest and to address the research question. The study followed the methodological principle of saturation. Saturation occurred as adding more participants to the study would not have resulted in additional perspectives or information regarding the papers core topic, i.e. the exploration whether scholars as government-appointed research evaluators create congruence between their professional quality standards and political demands. However, due to the qualitative small-N study design the findings are limited in terms of their breadth and scope. One major disadvantage of purposive samples like the one used in this study is that they can be prone to a researcher bias. The judgmental, subjective component of purposive sampling is a weakness as the sample has been based on the researcher’s judgment. Compared to probability sampling techniques designed to reduce researcher biases this is a clear disadvantage. However, the judgment concerning the purposive sample drawn on in this study has been based on clear theory-driven criteria that were illustrated in the data analysis section. The small N design and purposive sampling procedure have clear deficits concerning theoretical, logical, or analytical generalization. It’s worth investigating whether selecting different units leads to different results. Further studies could survey more evaluators from additional disciplines who took part in informed peer review procedures and validate the findings by using probability sampling techniques. In addition, it is worth investigating how structural changes in evaluation procedures, such as the time available for review, affect the behavior of reviewers and especially the core results of this study. This could provide policy makers with additional helpful information regarding the design of evaluation procedures.” (467-488)

Addition of Figure:

Figure 1. Data Analysis

Reviewer #2 comment on Results

“The findings are illuminating but it is worth further categorizing the themes and elaborating on them as it appears that there are a number of subsidiary aspects to each theme.

For example, for the theme of Motivation for participation: Evaluating as part of the professional role (line 229), the subsidiary aspects are:

• Duty to the academic profession

• Duty to the community

• Protecting research integrity

The same can be applied to other themes to provide more clarity to the reader.”

Authors’ response:

Thank you very much for your useful comment. I revised the manuscript in order to provide more clarity to the reader. The findings of each theme are now illustrated along subsidiary aspects. These aspects were already included in the data analysis and are now also used to arrange the findings more clearly in order to provide clarity to the reader. I have also added a graphic that illustrates the data analysis and includes the subsidiary aspects (see above). I have improved the text as shown below:

“Motivation for participation: Evaluating as part of the professional role

The data suggest that the motivation for participating in government-initiated research evaluations is predominantly driven by three central aspects: a perceived duty to the academic profession, a perceived duty to the public, and a will to protect the integrity of research.

Duty to the academic profession

The interviewed scholars consider participation in evaluations as a part of scientific self-administration and as a duty towards the academic profession. (…) V1 also regards evaluations as a commitment to his colleagues, when he says: “I also use peer review services” (V1: 36-37).

Duty to the public

The interviewees see it as their task to account to the public for the actions of scholars on behalf of science. (…) He emphasizes the positive effects of the transparency generated by evaluations: “In some cases civil servants’ research performance can be improved - to put it mildly” (V1: 59-63).

Protecting research integrity

The interviewees stress they do not accept externally defined criteria of high research quality and claim to be self-determined in judging research performance. (…) Acting as an evaluator during politically initiated evaluations prevents “influences from outside science gaining too much weight” (E2: 58-59).

Perception of politically initiated research evaluations

The data suggest that the interviewees are critical of the growing relevance of politically initiated research evaluations. This criticism is driven by two central aspects: a perceived threat to professional self-regulation and deficits in evaluation procedures.

Threat to professional self-regulation

Scholars perceive politically initiated evaluations as a threat to their professions’ core activities and their self-regulation. (…) “It is difficult to apply research funds with unconventional proposals, because other scholars read the proposal and say: ‘This is impossible’” (E1: 165-168). 

Deficits of evaluation procedures

Economist V2 criticizes the superficial character of evaluations: (…) With the exception of one interviewee, all of the interviewed evaluators share this critical stance towards output indicators for measuring research, since these are objectified by external actors and based on quantifying methods.” (243-281)

“Evaluating research quality in the role of a research evaluator

When describing the characteristics of high-quality research in the context of research evaluations, the evaluators of all three disciplines refer to similar criteria. There are no significant subject-specific differences with regard to the criteria applied. All interviewed evaluators refer to similar productivity indicators. 

Number and outlet of publications

Reviewers from all three disciplines state that they are mainly guided by the number of publications and – except for historians – journal rankings. (…) E1 emphasizes that international visibility always plays a role, “that is (...) how far things have been taken up internationally by other scholars, influenced their work and the like” (E1: 225-228).

Coherence

Reviewers from all three disciplines state that they were paying attention to the coherence of the research profile of the evaluated institutes, emphasizing, for example, it matters whether an institute had “a coherent program” (V2: 149) as it is a sign of quality if “there is some kind of center formation, so that one can also achieve something as a group” (V2: 150-151), (…)

Internationality

Reviewers in all three disciplines refer to internationality stressing, (…) or “care was taken to ensure that the faculties were then also internationally networked” (V1: 232-233).

Third-party funding

Interviewees emphasize the importance of third-party funding, which plays a role even though it is “more of a function derived from something else” (V2: 152-153). In electrical engineering, spin-offs of companies and patents play a role.” (350-385)

Addition of Figure:

Figure 1. Data Analysis (line 239)

Reviewer #2 comment on Discussion and Conclusion

“Is there any published literature to support the conclusions of the study – lines 410-423?

Line 422 - …it should be “research performance…” rather than performances.

Line 427 – “unconventional research approaches that not succeed in terms of…”

should be “unconventional research approaches that do not succeed in terms of…” Please revise the language accordingly.

More practical and theoretical implications of the findings should be given.

The limitations of this study should be elaborated on, with the main one being the small sample size and that qualitative studies have limitations in terms of their breath and scope.

More suggestions for further research should be provided.”

Authors’ response:

Thank you very much for your useful comment. I revised the manuscript accordingly. I deleted the argumentation of line 410-423 (of the initial document) and rearranged the discussion and conclusion section. I revised the language of both text passages in line 422 and 427 (of the initial document) according to your remark. I elaborated more extensively on the implications and limitations of this study, especially the small sample size and the limitations qualitative studies have in terms of their breadth and scope. I have improved the text as shown below:

“The study has several limitations that create necessity for further research. The sample size was large enough to sufficiently explore the phenomenon of interest and to address the research question. The study followed the methodological principle of saturation. Saturation occurred as adding more participants to the study would not have resulted in additional perspectives or information regarding the papers core topic, i.e. the exploration whether scholars as government-appointed research evaluators create congruence between their professional quality standards and political demands. However, due to the qualitative small-N study design the findings are limited in terms of their breadth and scope. One major disadvantage of purposive samples like the one used in this study is that they can be prone to a researcher bias. The judgmental, subjective component of purposive sampling is a weakness as the sample has been based on the researcher’s judgment. Compared to probability sampling techniques designed to reduce researcher biases this is a clear disadvantage. However, the judgment concerning the purposive sample drawn on in this study has been based on clear theory-driven criteria that were illustrated in the data analysis section. The small N design and purposive sampling procedure have clear deficits concerning theoretical, logical, or analytical generalization. It’s worth investigating whether selecting different units leads to different results. Further studies could survey more evaluators from additional disciplines who took part in informed peer review procedures and validate the findings by using probability sampling techniques. In addition, it is worth investigating how structural changes in evaluation procedures, such as the time available for review, affect the behavior of reviewers and especially the core results of this study. This could provide policy makers with additional helpful information regarding the design of evaluation procedures.” (467-488)

---

## [Editor Report · Decision Letter 1]

4 Sep 2020

Scholars as government-appointed research evaluators: Do they create congruence between their professional quality standards and political demands?

PONE-D-20-07099R1

Dear Dr. Woiwode,

We’re pleased to inform you that your manuscript has been judged scientifically suitable for publication and will be formally accepted for publication once it meets all outstanding technical requirements.

Kind regards,

Cindy Sing-bik Ngai

Academic Editor

PLOS ONE

Additional Editor Comments (optional):

I appreciate the author spent time responding to reviewers' comments and revising the manuscript, especially the inclusion of Figure 1 which provides clear elaboration on the coding procedure in conducting qualitative content analysis.
---

## [Editor Report · Acceptance letter]

30 Sep 2020

PONE-D-20-07099R1 

Scholars as government-appointed research evaluators: Do they create congruence between their professional quality standards and political demands? 

Dear Dr. Woiwode:

I'm pleased to inform you that your manuscript has been deemed suitable for publication in PLOS ONE. Congratulations! Your manuscript is now with our production department. 

Kind regards, 

on behalf of

Dr. Cindy Sing Bik Ngai 

Academic Editor

PLOS ONE